# Capacity Bounded Differential Privacy

**Kamalika Chaudhuri**
UC San Diego
kamalika@cs.ucsd.edu

**Jacob Imola**
UC San Diego
jimola@eng.ucsd.edu

**Ashwin Machanavajjhala**
Duke University
ashwin@cs.duke.edu

## Abstract

Differential privacy has emerged as the gold standard for measuring the risk posed by an algorithm's output to the privacy of a single individual in a dataset. It is defined as the worst-case distance between the output distributions of an algorithm that is run on inputs that differ by a single person. In this work, we present a novel relaxation of differential privacy, *capacity bounded differential privacy*, where the adversary that distinguishes the output distributions is assumed to be *capacity-bounded* – i.e. bounded not in computational power, but in terms of the function class from which their attack algorithm is drawn. We model adversaries of this form using restricted $f$-divergences between probability distributions, and study properties of the definition and algorithms that satisfy them. Our results demonstrate that these definitions possess a number of interesting properties enjoyed by differential privacy and some of its existing relaxations; additionally, common mechanisms such as the Laplace and Gaussian mechanisms enjoy better privacy guarantees for the same added noise under these definitions.

## 1 Introduction

Differential privacy [8] has emerged as a gold standard for measuring the privacy risk posed by algorithms analyzing sensitive data. A randomized algorithm satisfies differential privacy if an arbitrarily powerful attacker is unable to distinguish between the output distributions of the algorithm when the inputs are two datasets that differ in the private value of a single person. This provides a guarantee that the additional disclosure risk to a single person in the data posed by a differentially private algorithm is limited, even if the attacker has access to side information. However, a body of prior work [28, 3, 17, 1] has shown that this strong privacy guarantee comes at a cost: for many machine-learning tasks, differentially private algorithms require a much higher number of samples to acheive the same amount of accuracy than is needed without privacy.

Prior work has considered relaxing differential privacy in a number of different ways. Pufferfish [16] and Blowfish [12] generalize differential privacy by restricting the properties of an individual that should not be inferred by the attacker, as well as explicitly enumerating the side information available to the adversary. Renyi- and KL-differential privacy [23, 31] measure privacy loss as the $\alpha$-Renyi and KL-divergence between the output distributions (respectively). The original differential privacy definition measures privacy as a max-divergence (or $\alpha$-Renyi, with $\alpha \to \infty$). Computational differential privacy (CDP) [24] considers a computationally bounded attacker, and aims to ensure that the output distributions are computationally indistinguishable. These three approaches are orthogonal to one another as they generalize or relax different aspects of the privacy definition.

In this paper, we consider an novel approach to relaxing differential privacy by restricting the adversary to "attack" or post-process the output of a private algorithm using functions drawn from a *restricted function class* and show how to quantitatively calculate privacy losses against particular function classes. These adversaries, that we call *capacity bounded*, can be used to model two kinds of application scenarios. The first is where the attacker is machine learnt and lies in some known space of functions (e.g., all linear functions, linear classifiers, etc.). The second is a user under a data-usage

contract that restricts how the output of a private algorithm can be used. If the contract stipulates that the user can only compute a certain class of functions on the output, then a privacy guarantee of this form ensures that no privacy violation can occur if users obey their contracts. By showing how to quatify privacy loss in these settings allows (a) better decisions in cases where we expect the adversaries to be bounded in what they can do – for example, automated adversaries or adversaries under a data-usage contract – and (b) better design of data-usage contracts. Unlike computational DP, where computationally bounded adversaries do not meaningfully relax the privacy definition in the typical centralized differential privacy model [11], we believe that capacity bounded adversaries will relax the definition to permit more useful algorithms and are a natural and interesting class of adversaries.

The first challenge is how to model these adversaries. We begin by showing that privacy with capacity bounded adversaries can be cleanly modeled through the restricted divergences framework [21, 20, 26] that has been recently used to build a theory for generative adversarial networks. This gives us a notion of $(\mathcal{H}, \Gamma)$-*capacity bounded differential privacy*, where the privacy loss is measured in terms of a divergence $\Gamma$ (e.g., Renyi) between output distributions of a mechanism on datasets that differ by a single person restricted to functions in $\mathcal{H}$ (e.g., $lin$, the space of all linear functions).

We next investigate properties of these privacy definitions, and show that they enjoy many of the good properties enjoyed by differential privacy and its relaxations – convexity, graceful composition, as well as post-processing invariance to certain classes of functions. We analyze well-known privacy mechanisms, such as the Laplace and the Gaussian mechanism under $(lin, \mathbb{KL})$ and $(lin, Renyi)$ capacity bounded privacy – where the adversaries are the class of all linear functions. We show that restricting the capacity of the adversary does provide improvements in the privacy guarantee in many cases. We then use this to demonstrate that the popular Matrix Mechanism [18, 19, 22] gives an improvement in the privacy guarantees when considered under capacity bounded definition.

We conclude by showing some preliminary results that indicate that the capacity bounded definitions satisfy a form of algorithmic generalization. Specifically, for every class of queries $\mathcal{Q}$, there exists a (non-trivial) $\mathcal{H}$ such that an algorithm that answers queries in the class $\mathcal{Q}$ and is $(\mathcal{H}, \mathbb{KL})$-capacity bounded private with parameter $\epsilon$ also ensures generalization with parameter $O(\sqrt{\epsilon})$.

The main technical challenge we face is that little is known about properties of restricted divergences. While unrestricted divergences such as KL and Renyi are now well-understood as a result of more than fifty years of research in information theory, these restricted divergences are only beginning to be studied in their own right. A side-effect of our work is that we advance the information geometry of these divergences, by establishing properties such as versions of Pinsker's Inequality and the Data Processing Inequality. We believe that these will be of independent interest to the community and aid the development of the theory of GANs, where these divergences are also used.

## 2 Preliminaries

### 2.1 Privacy

Let $D$ be a dataset, where each data point represents a single person's value. A randomized algorithm $A$ satisfies differential privacy [8] if its output is insensitive to adding or removing a data point to its input $D$. We can define this privacy notion in terms of the Renyi Divergence of two output distributions: $A(D)$ – the distribution of outputs generated by $A$ with input $D$, and $A(D')$, the distrbution of outputs generated by $A$ with input $D'$, where $D$ and $D'$ differ by a single person's value [23]. Here, recall that the Renyi divergence of order $\alpha$ between distributions $P$ and $Q$ can be written as: $\mathcal{D}_{R,\alpha}(P, Q) = \frac{1}{\alpha-1} \log \left( \int_x P(x)^\alpha Q(x)^{1-\alpha} dx \right)$.

**Definition 1** (Renyi Differential Privacy). *A randomized algorithm $A$ that operates on a dataset $D$ is said to provide $(\alpha, \epsilon)$-Renyi differential privacy if for all $D$ and $D'$ that differ by a single person's value, we have: $\mathcal{D}_{R,\alpha}(A(D), A(D')) \leq \epsilon$.*

When the order of the divergence $\alpha \to \infty$, we require the max-divergence of the two distrbutions bounded by $\epsilon$ – which is standard differential privacy [7]. When $\alpha \to 1$, $\mathcal{D}_{R,\alpha}$ becomes the Kullback-Liebler (KL) divergence, and we get KL differential privacy [32].

## 2.2 Divergences and their Variational Forms

A popular class of divergences is Czisar's $f$-divergences [5], defined as follows.

**Definition 2.** *Let $f$ be a lower semi-continuous convex function such that $f(1) = 0$, and let $P$ and $Q$ be two distributions over a probability space $(\Omega, \Sigma)$ such that $P$ is absolutely continuous with respect to $Q$. Then, the $f$-divergence between $P$ and $Q$, denoted by $\mathcal{D}_f(P, Q)$ is defined as: $\mathcal{D}_f(P, Q) = \int_\Omega f\left(\frac{dP}{dQ}\right) dQ$.*

Examples of $f$-divergences include the KL divergence ($f(t) = t \log t$), the total variation distance ($f(t) = \frac{1}{2}|t - 1|$) and $\alpha$-divergence ($f(t) = (|t|^\alpha - 1)/(\alpha^2 - \alpha)$).

Given a function $f$ with domain $\mathbf{R}$, we use $f^*$ to denote its Fenchel conjugate: $f^*(s) = \sup_{x \in \mathbf{R}} x \cdot s - f(x)$. [25] shows that $f$-divergences have a dual variational form:

$$\mathcal{D}_f(P, Q) = \sup_{h \in \mathcal{F}} \mathbb{E}_{x \sim P}[h(x)] - \mathbb{E}_{x \sim Q}[f^*(h(x))], \tag{1}$$

where $\mathcal{F}$ is the set of all functions over the domain of $P$ and $Q$.

**Restricted Divergences.**  Given an $f$-divergence and a class of functions $\mathcal{H} \subseteq \mathcal{F}$, we can define a notion of a $\mathcal{H}$-restricted $f$-divergence by selecting, instead of $\mathcal{F}$, the more restricted class of functions $\mathcal{H}$, to maximize over in (1):

$$\mathcal{D}_f^{\mathcal{H}}(P, Q) = \sup_{h \in \mathcal{H}} \mathbb{E}_{x \sim P}[h(x)] - \mathbb{E}_{x \sim Q}[f^*(h(x))], \tag{2}$$

These restricted divergences have previously been considered in the context of, for example, Generative Adversarial Networks [26, 2, 20, 21].

While Renyi divergences are not $f$-divergences in general, we can also define restricted versions for them by going through the corresponding $\alpha$-divergence – which, recall, is an $f$-divergence with $f(t) = (|t|^\alpha - 1)/(\alpha^2 - \alpha)$, and is related to the Renyi divergence by a closed form equation [4]. Given a function class $\mathcal{H}$, an order $\alpha$, and two probability distributions $P$ and $Q$, we can define the $\mathcal{H}$-restricted Renyi divergence of order $\alpha$ using the same closed form equation on the $\mathcal{H}$-restricted $\alpha$-divergence as follows:

$$\mathcal{D}_{R,\alpha}^{\mathcal{H}}(P, Q) = \left(\log\left(1 + \alpha(\alpha - 1)\mathcal{D}_\alpha^{\mathcal{H}}(P, Q)\right)\right)/(\alpha - 1) \tag{3}$$

where $\mathcal{D}_\alpha^{\mathcal{H}}$ is the corresponding $\mathcal{H}$-restricted $\alpha$-divergence.

## 3  Capacity Bounded Differential Privacy

The existence of $\mathcal{H}$-restricted divergences suggests a natural notion of privacy – when the adversary lies in a (restricted) function class $\mathcal{H}$, we can, instead of $\mathcal{F}$, consider the class $\mathcal{H}$ of functions in the supremum. This enforces that no adversary in the function class $\mathcal{H}$ can distinguish between $A(D)$ and $A(D')$ beyond $\epsilon$. We call these *capacity bounded adversaries*.

**Definition 3** (($\mathcal{H}, \Gamma$)-Capacity Bounded Differential Privacy). *Let $\mathcal{H}$ be a class of functions with domain $\mathcal{X}$, and $\Gamma$ be a divergence. A mechanism $A$ is said to offer $(\mathcal{H}, \Gamma)$-capacity bounded privacy with parameter $\epsilon$ if for any two $D$ and $D'$ that differ by a single person's value, the $\mathcal{H}$-restricted $\Gamma$-divergence between $A(D)$ and $A(D')$ is at most $\epsilon$:*

$$\Gamma^{\mathcal{H}}(A(D), A(D')) \leq \epsilon$$

When $\mathcal{H}$ is the class of all functions, and $\Gamma$ is a Renyi divergence, the definition reduces to Renyi Differential privacy; capacity bounded privacy is thus a generalization of Renyi differential privacy.

**Function Classes.**  The definition of capacity bounded privacy allows for an infinite number of variations corresponding to the class of adversaries $\mathcal{H}$.

An example of such a class is all linear adversaries over a feature space $\phi$, which includes all linear regressors over $\phi$. A second example is the class of all functions in an Reproducible Kernel Hilbert Space; these correspond to all kernel classifiers. A third interesting class is linear combinations of all Relu functions; this correspond to all two layer neural networks. These function classes would capture typical machine learnt adversaries, and designing mechanisms that satisfy capacity bounded DP with respect to these functions classes is an interesting research direction.

# 4 Properties

The success of differential privacy has been attributed its highly desirable properties that make it amenable for practical use. In particular, [15] proposes that any privacy definition should have three properties – convexity, post-processing invariance and graceful composition – all of which apply to differential privacy. We now show that many of these properties continue to hold for the capacity bounded definitions. The proofs appear in the Appendix.

**Post-processing.** Most notions of differential privacy satisfy post-processing invariance, which states that applying any function to the output of a private mechanism does not degrade the privacy guarantee. We cannot expect post-processing invariance to hold with respect to all functions for capacity bounded privacy – otherwise, the definition would be equivalent to privacy for all adversaries!

However, we can show that for any $\mathcal{H}$ and for any $\Gamma$, $(\mathcal{H}, \Gamma)$-capacity bounded differential privacy is preserved after post-processing if certain conditions about the function classes hold:

**Theorem 1.** *Let $\Gamma$ be an $f$-divergence or the Renyi divergence of order $\alpha > 1$, and let $\mathcal{H}$ $\mathcal{G}$, and $\mathcal{I}$ be function classes such that for any $g \in \mathcal{G}$ and $i \in \mathcal{I}$, $i \circ g \in \mathcal{H}$. If algorithm $A$ satisfies $(\mathcal{H}, \Gamma)$-capacity bounded privacy with parameter $\epsilon$, then, for any $g \in \mathcal{G}$, $g \circ A$ satisfies $(\mathcal{I}, \Gamma)$-capacity bounded privacy with parameter $\epsilon$.*

Specifically, if $\mathcal{I} = \mathcal{H}$, then $A$ is post-processing invariant. Theorem 1 is essentially a form of the popular Data Processing Inequality applied to restricted divergences; its proof is in the Appendix and follows from the definition as well as algebra. An example of function classes $\mathcal{G}, \mathcal{H}$, and $\mathcal{I}$ that satisfy this conditions is when $\mathcal{G}, \mathcal{H}, \mathcal{I}$ are linear functions, where $\mathcal{G} : \mathbf{R}^s \to \mathbf{R}^d$, $\mathcal{H} : \mathbf{R}^s \to \mathbf{R}$, and $\mathcal{I} : \mathbf{R}^d \to \mathbf{R}$.

**Convexity.** A second property is convexity [14], which states that if $A$ and $B$ are private mechanisms with privacy parameter $\epsilon$ then so is a composite mechanism $M$ that tosses a (data-independent) coin and chooses to run $A$ with probability $p$ and $B$ with probability $1 - p$. We show that convexity holds for $(\mathcal{H}, \Gamma)$-capacity bounded privacy for any $\mathcal{H}$ and any $f$-divergence $\Gamma$.

**Theorem 2.** *Let $\Gamma$ be an $f$-divergence and $A$ and $B$ be two mechanisms which have the same range and provide $(\mathcal{H}, \Gamma)$-capacity bounded privacy with parameter $\epsilon$. Let $M$ be a mechanism which tosses an independent coin, and then executes mechanism $A$ with probability $\lambda$ and $B$ with probability $1 - \lambda$. Then, $M$ satisfies $(\mathcal{H}, \Gamma)$-capacity bounded privacy with parameter $\epsilon$.*

We remark that while differential privacy and KL differential privacy satisfy convexity, (standard) Renyi differential privacy does not; it is not surprising that neither does its capacity bounded version. The proof uses convexity of the function $f$ in an $f$-divergence.

**Composition.** Broadly speaking, composition refers to how privacy properties of algorithms applied multiple times relate to privacy properties of the individual algorithms. Two styles of composition are usually considered – sequential and parallel.

A privacy definition is said to satisfy *parallel composition* if the privacy loss obtained by applying multiple algorithms on disjoint datasets is the maximum of the privacy losses of the individual algorithms. In particular, Renyi differential privacy of any order satisfies parallel composition. We show below that so does capacity bounded privacy.

**Theorem 3.** *Let $\mathcal{H}_1, \mathcal{H}_2$ be two function classes that are convex and translation invariant. Let $\mathcal{H}$ be the function class:*

$$\mathcal{H} = \{h_1 + h_2 | h_1 \in \mathcal{H}_1, h_2 \in \mathcal{H}_2\}$$

*and let $\Gamma$ be the KL divergence or the Renyi divergence of order $\alpha > 1$. If mechanisms $A$ and $B$ satisfy $(\mathcal{H}_1, \Gamma)$ and $(\mathcal{H}_2, \Gamma)$ capacity bounded privacy with parameters $\epsilon_1$ and $\epsilon_2$ respectively, and if the datasets $D_1$ and $D_2$ are disjoint, then the combined release $(A(D_1), B(D_2))$ satisfies $(\mathcal{H}, \Gamma)$ capacity bounded privacy with parameter $\max(\epsilon_1, \epsilon_2)$.*

In contrast, a privacy definition is said to compose *sequentially* if the privacy properties of algorithms that satisfy it degrade gracefully as the same dataset is used in multiple private releases. In particular, Renyi differential privacy is said to satisfy sequential additive composition – if multiple private algorithms are used on the same dataset, then their privacy parameters add up. We show below that

| Divergence | Mechanism | Privacy Parameter, Linear Adversary | Privacy Parameter, Unrestricted |
|---|---|---|---|
| KL | Laplace | $\sqrt{1+\epsilon^2} - 1 + \log\left(1 - \frac{(\sqrt{1+\epsilon^2}-1)^2}{\epsilon^2}\right)$ | $\epsilon - 1 + e^{-\epsilon}$ |
| KL | Gaussian | $1/2\sigma^2$ | $1/2\sigma^2$ |
| $\alpha$-Renyi | Laplace | $\leq \frac{1}{\alpha-1}\log(1 + 2^{\alpha-1}\epsilon^\alpha)$ | $\geq \epsilon - \log(2)/\alpha-1$ |
| $\alpha$-Renyi | Gaussian | $\leq \frac{1}{\alpha-1}\log(1 + \sqrt{2\pi}^{\alpha-1}/\sigma^\alpha)$ | $\alpha/2\sigma^2$ |
| $\alpha$-Renyi | Laplace, $d$-dim | $\leq \frac{1}{\alpha-1}\log(1 + 2^{d(\alpha-1)}(\epsilon\|v\|_\alpha)^\alpha)$ | $\geq \epsilon\|v\|_1 - d\log(2)/\alpha-1$ |
| $\alpha$-Renyi | Gaussian, $d$-dim | $\leq \frac{1}{\alpha-1}\log\left(1 + \frac{2^{d(\alpha-1)}\sqrt{\pi/2}^{\alpha-1}\|v\|_\alpha^\alpha}{\sigma^\alpha}\right)$ | $\frac{\alpha\|v\|_2^2}{2\sigma^2}$ |

Table 1: Privacy parameters of different mechanisms and divergences with a linear adversary and unrestricted. Proofs appear in the Appendix.

a similar result can be shown for $(\mathcal{H}, \Gamma)$-capacity bounded privacy when $\Gamma$ is the KL or the Renyi divergence, and $\mathcal{H}$ satisfies some mild conditions.

**Theorem 4.** *Let $\mathcal{H}_1$ and $\mathcal{H}_2$ be two function classes that are convex, translation invariant, and that includes a constant function. Let $\mathcal{H}$ be the function class:*

$$\mathcal{H} = \{h_1 + h_2 | h_1 \in \mathcal{H}_1, h_2 \in \mathcal{H}_2\}$$

*and let $\Gamma$ be the KL divergence or the Renyi divergence of order $\alpha > 1$. If mechanisms $A$ and $B$ satisfy $(\mathcal{H}_1, \Gamma)$ and $(\mathcal{H}_2, \Gamma)$ capacity bounded privacy with parameters $\epsilon_1$ and $\epsilon_2$ respectively, then the combined release $(A, B)$ satisfies $(\mathcal{H}, \Gamma)$ capacity bounded privacy with parameter $\epsilon_1 + \epsilon_2$.*

The proof relies heavily on the relationship between the restricted and unrestricted divergences, as shown in [21, 20, 9], and is provided in the Appendix. Observe that the conditions on $\mathcal{H}_1$ and $\mathcal{H}_2$ are rather mild, and include a large number of interesting functions. One such example of $\mathcal{H}$ is the set of ReLU neural networks with linear output node, a common choice when performing neural network regression.

The composition guarantees offered by Theorem 4 are non-adaptive – the mechanisms $A$ and $B$ are known in advance, and $B$ is not chosen as a function of the output of $A$. Whether fully general adaptive composition is possible for the capacity bounded definitions is left as an open question for future work.

## 5 Privacy Mechanisms

The definition of capacity bounded privacy allows for an infinite number of variations, corresponding to the class of adversaries $\mathcal{H}$ and divergences $\Gamma$, exploring all of which is outside the scope of a single paper. For the sake of concreteness, we consider *linear* and (low-degree) *polynomial* adversaries for $\mathcal{H}$ and KL as well as Renyi divergences of order $\alpha$ for $\gamma$. These correspond to cases where a linear or a low-degree polynomial function is used by an adversary to attack privacy.

A first sanity check is to see what kind of linear or polynomial guarantee is offered by a mechanism that directly releases a non-private value (without any added randomness). This mechanism offers no finite linear KL or Renyi differential privacy parameter – which is to be expected from any sensible privacy definition (see the Appendix).

We now look at the capacity bounded privacy properties of the familiar Laplace and Gaussian mechanisms which form the building blocks for much of differential privacy. Bounds we wish to compare appear in Table 1.

**Laplace Mechanism.** The Laplace mechanism adds $Lap(0, 1/\epsilon)$ noise to a function with global sensitivity 1. In $d$ dimensions, the mechanism adds $d$ i.i.d. samples from $Lap(0, 1/\epsilon)$ to a function with $L_1$ sensitivity 1. More generally, we consider functions whose global sensitivity along coordinate $i$ is $v_i$. We let $v = (v_1, v_2, \ldots, v_d)$.

Table 1 shows $(lin, KL)$-capacity bounded privacy and KL-DP parameters for the Laplace mechanism. The former has a slightly smaller parameter than the latter.

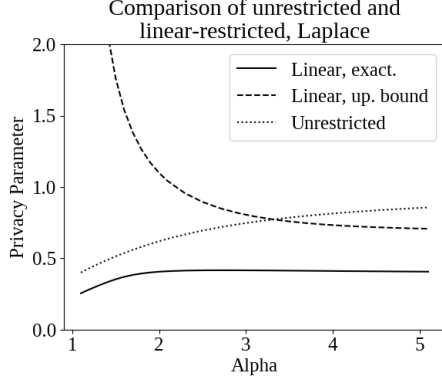 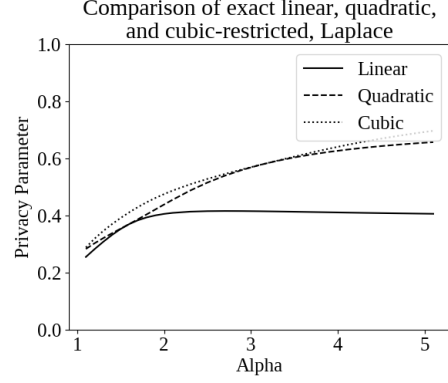

(a) Plots of $(lin, Renyi)$ capacity bounded DP and Renyi-DP parameters for Laplace mechanism when $\epsilon = 1$. For $(lin, Renyi)$, the upper bound and exact value are shown.

(b) Comparison of exact values of $(poly, Renyi)$ capacity bounded DP parameters for Laplace mechanism when $\epsilon = 1$.

Table 1 also contains an upper bound on the $(lin, Renyi)$ capacity bounded privacy, and a lower bound on the Renyi-DP. The exact value of the Renyi-DP is:

$$\frac{1}{\alpha - 1} \log \left( \left( \frac{1}{2} + \frac{1}{4\alpha - 2} \right) e^{(\alpha-1)\epsilon} + \left( \frac{1}{2} - \frac{1}{4\alpha - 2} \right) e^{-\alpha\epsilon} \right) \tag{4}$$

By multiplying by $\alpha - 1$ and exponentiating, we see that the $(lin, Renyi)$ upper bound grows with $1 + \epsilon(2\epsilon)^{\alpha-1}$, while the Renyi-DP lower bound grows with $(e^\epsilon)^{\alpha-1}$. This means no matter what $\epsilon$ is, a moderately-sized $\alpha$ will make the $(lin, Renyi)$ upper bound smaller than the Renyi lower bound.

Figure 1a plots the $(lin, Renyi)$ upper bound, (4), and the exact value of the $(lin, Renyi)$ parameter, as functions of $\alpha$ when $\epsilon = 1$. We see the exact $(lin, Renyi)$ is always better than (4), although the upper bound may sometimes be worse. The upper bound overtakes the lower bound when $\alpha \approx 3.3$.

For the multidimensional Laplace Mechanism, the story is the same. The $(lin, Renyi)$ upper bound can now be thought of as a function of $\epsilon\|v\|_\alpha$, and the $Renyi$ lower bound a function of $\epsilon\|v\|_1$. Because $\|v\|_\alpha \leq \|v\|_1$, we can replace $\|v\|_\alpha$ with $\|v\|_1$ in the $(lin, Renyi)$ upper bound, and repeat the analysis for the unidimensional case. Notice that our $(lin, Renyi)$ upper bound is slightly better than using composition $d$ times on the unidimensional Laplace mechanism which would result in a multiplicative factor of $d$.

Figure 1b contains plots of the exact $(poly, Renyi)$ paramters for degree 1,2, and 3 polynomials, as functions of $\alpha$ when $\epsilon = 1$. As we expect, as the polynomial complexity increases, the $(poly, Renyi)$ parameters converge to the Renyi-DP parameter. This also provides an explanation for the counterintuitive observation that the $(poly, Renyi)$ parameters eventually decrease with $\alpha$. The polynomial function classes are too simple to distinguish the two distributions for larger $\alpha$, but their ability to do so increases as the polynomial complexity increases.

**Gaussian Mechanism.** The Gaussian mechanism adds $\mathcal{N}(0, \sigma^2)$ noise to a function with global sensitivity 1. In $d$ dimensions, the mechanism adds $\mathcal{N}(0, \sigma^2 I_d)$ to a function with $L_2$ sensitivity 1. More generally, we consider functions whose global sensitivity along coordinate $i$ is $v_i$ We let $v = (v_1, v_2, \ldots, v_d)$.

Whereas the $(lin, \mathbb{KL})$ parameter for Laplace is a little better than the KL-DP parameter, Table 1 shows the Gaussian mechanism has the same parameter. This is because if $P$ and $Q$ are two Gaussians with equal variance, the function $h$ that maximizes the variational formulation corresponding to the KL-divergence is a linear function.

For Renyi capacity bounded privacy, the observations we make are nearly identical to that of the Laplace Mechanism. The reader is referred to the Appendix for plots and specific details.

**Matrix Mechanism.**

Now, we show how to use the bounds in Table 1 to obtain better capacity bounded parameters for a composite mechanism often used in practice: the Matrix mechanism [18, 19, 22]. The Matrix mechanism is a very general method of computing linear queries on a dataset, usually with less error than the Laplace Mechanism. Given a dataset $D \in \Sigma^m$ over a finite alphabet $\Sigma$ of size $n$, we can form a vector of counts $x \in \mathbf{R}^n$ such that $x_i$ contains how many times $i$ appears in $D$. A linear query is a vector $w \in \mathbf{R}^n$ and has answer $w^T x$. A set of $d$ linear queries can then be given by a matrix $W \in \mathbf{R}^{d \times n}$ with the goal of computing $Wx$ privately.

A naive way to do this is to use the Laplace Mechanism in $d$ dimensions to release $x$ and then multiply by $W$. The key insight is that, for any $A \in \mathbf{R}^{s \times n}$ of rank $n$, we can instead add noise to $Ax$ and multiply the result by $WA^\dagger$. Here, $A^\dagger$ denotes the pseudoinverse of $A$ such that $WA^\dagger A = W$.

The Laplace Mechanism arises as the special case $A = I$; however, more carefully chosen $A$s may be used to get privacy with less noise. This gives rise to the Matrix mechanism:

$$M_A(W, x, \epsilon) = WA^\dagger(Ax + \|A\|_1 Lap_d(0, 1/\epsilon))$$

Here, $\|A\|_1$ is the maximum $L_1$-norm of any column of $a$. Prior work shows that this mechanism provides differential privacy and suggest different methods for picking an $A$. Regardless of which $A$ is chosen, , we are able to provide a capacity-bounded privacy parameter that is better than any known Renyi-DP analysis has shown:

**Theorem 5** (Matrix Mechanism). *Let $x \in \mathbf{R}^n$ be a data vector, $W \in \mathbf{R}^{d \times n}$ be a query matrix, and $A \in \mathbf{R}^{s \times n}$ be a strategy matrix. Then, releasing $M_A(W, x, \epsilon)$ offers $(lin, Renyi)$ capacity bounded privacy with parameter at most $\frac{1}{\alpha-1} \log(1 + 2^{s(\alpha-1)}\epsilon^\alpha)$.*

Note this is the same upper bound as the $d$-dimensional Laplace mechanism; indeed, the proof works by applying post-processing to the Laplace mechanism.

## 6 Algorithmic Generalization

Overfitting to input data has long been the curse of many statistical and machine learning methods; harmful effects of overfitting can range from poor performance at deployment time all the way up to lack of reproducibility in scientific research due to $p$-hacking [13]. Motivated by these concerns, a recent line of work in machine learning investigates properties that algorithms and methods should possess so that they can automatically guarantee generalization [27, 10, 6, 29]. In this connection, differential privacy and many of its relaxations have been shown to be highly successful; it is known for example, that if adaptive data analysis is done by a differentially private algorithm, then the results automatically possess certain generalization guarantees.

A natural question is whether these properties translate to the capacity bounded differential privacy, and if so, under what conditions. We next investigate this question, and show that capacity bounded privacy does offer promise in this regard. A more detailed investigation is left for future work.

**Problem Setting.**    More specifically, the problem setting is as follows. [27, 10, 6, 29]. We are given as input a data set $S = \{x_1, \ldots, x_n\}$ drawn from an (unknown) underlying distribution $D$ over an instance space $\mathcal{X}$, and a set of "statistical queries" $\mathcal{Q}$; each statistical query $q \in \mathcal{Q}$ is a function $q : \mathcal{X} \to [0, 1]$.

A data analyst $M$ observes $S$, and then picks a query $q_S \in \mathcal{Q}$ based on her observation; we say that $M$ *generalizes well* if the query $q_S$ evaluated on $S$ is close to $q_S$ evaluated on a fresh sample from $D$ (on expectation); more formally, this happens when the *generalization gap* $\frac{1}{n}\sum_{i=1}^n q_S(x_i) - \mathbb{E}_{x \sim D}[q_S(x)]$ is low.

Observe that if the query was picked without an apriori look at the data $S$, then the problem would be trivial and solved by a simple Chernoff bound. Thus bounding the generalization gap is challenging because the choice of $q_S$ depends on $S$, and the difficulty lies in analyzing the behaviour of particular methods that make this choice.

**Our Result.**    Prior work in generalization theory [27, 10, 6, 29] shows that if $M$ possesses certain algorithmic stability properties – such as differential privacy as well as many of its relaxations and

generalizations – then the gap is low. We next show that provided the adversarial function class $\mathcal{H}$ satisfies certain properties with respect to the statistical query class $\mathcal{Q}$, $(\mathcal{H}, lin)$-capacity bounded privacy also has good generalization properties.

**Theorem 6** (Algorithmic Generalization). *Let $S$ be a sample of size $n$ drawn from an underlying data distribution $D$ over an instance space $\mathcal{X}$, and let $M$ be a (randomized) mechanism that takes as input $S$, and outputs a query $q_S$ in a class $\mathcal{Q}$. For any $x \in \mathcal{X}$, define a function $h_x : \mathcal{Q} \to [0, 1]$ as: $h_x(q) = q(x)$, and let $\mathcal{H}$ be any class of functions that includes $\{h_x | x \in \mathcal{X}\}$.*

*If the mechanism $M$ satisfies $(\mathcal{H}, \mathbb{KL})$-capacity bounded privacy with parameter $\epsilon$, then, for every distribution $D$, we have:* $\left| \mathbb{E}_{S \sim D, M} \left( \frac{1}{n} \sum_{i=1}^{n} q_S(x_i) - \mathbb{E}_{x \sim D}[q_S(x)] \right) \right| \leq 8\sqrt{\epsilon}$.

We remark that the result would not hold for arbitrary $(\mathcal{H}, \mathbb{KL})$-capacity bounded privacy, and a condition that connects $\mathcal{H}$ to $\mathcal{Q}$ appears to be necessary. However, for specific distributions $D$, fewer conditions may be needed.

Observe also that Theorem 6 only provides a bound on expectation. Stronger guarantees – such as high probability bounds as well as adaptive generalization bounds – are also known in the adaptive data analysis literature. While we believe similar bounds should be possible in our setting, proving them requires a variery of information-theoretic properties of the corresponding divergences, which are currently not available for restricted divergences. We leave a deeper investigation for future work.

**Proof Ingredient: A Novel Pinsker-like Inequality.** We remark that an ingredient in the proof of Theorem 6 is a novel Pinsker-like inequality for restricted KL divergences, which was previously unknown, and is presented below (Theorem 7). We believe that this theorem may be of independent interest, and may find applications in the theory of generative adversarial networks, where restricted divergences are also used.

We begin by defining an integral probability metric (IPM) [30] with respect to a function class $\mathcal{H}$.

**Definition 4.** *Given a function class $\mathcal{H}$, and any two distributions $P$ and $Q$, the Integral Probability Metric (IPM) with respect to $\mathcal{H}$ is defined as follows: $IPM^{\mathcal{H}}(P, Q) = \sup_{h \in \mathcal{H}} |\mathbb{E}_P[h(x)] - \mathbb{E}_Q[h(x)]|$.*

Examples of IPMs include the total variation distance where $\mathcal{H}$ is the class of all functions with range $[0, 1]$, and the Wasserstein distance where $\mathcal{H}$ is the class of all $1$-Lipschitz functions. With this definition in hand, we can now state our result.

**Theorem 7** (Pinsker-like Inequality for Restricted KL Divergences). *Let $\mathcal{H}$ be a convex class of functions with range $[-1, 1]$ that is translation invariant and closed under negation. Then, for any $P$ and $Q$ such that $P$ is absolutely continuous with respect to $Q$, we have that: $IPM^{\mathcal{H}}(P, Q) \leq 8 \cdot \sqrt{\mathbb{KL}^{\mathcal{H}}(P, Q)}$.*

This theorem is an extended version of the Pinsker Inequality, which states that the total variation distance $TV(P, Q) \leq \sqrt{2\mathbb{KL}(P, Q)}$; however, instead of connecting the total variation distance and KL divergences, it connects IPMs and the corresponding restricted KL divergences.

# 7 Conclusion

We initiate a study into capacity bounded differential privacy – a relaxation of differential privacy against adversaries in restricted function classes. We show how to model these adversaries cleanly through the recent framework of restricted divergences. We then show that the definition satisfies privacy axioms, and permits mechanisms that have higher utility (for the same level of privacy) than regular KL or Renyi differential privacy when the adverary is limited to linear functions. Finally, we show some preliminary results that indicate that these definitions offer good generalization guarantees.

There are many future directions. A deeper investigation into novel mechanisms that satisfy the definitions, particularly for other function classes such as threshold and relu functions remain open. A second question is a more detailed investigation into statistical generalization – such as generalization in high probability and adaptivity – induced by these notions. Finally, our work motivates a deeper exploration into the information geometry of adversarial divergences, which is of wider interest to the community.

**Acknowledgments.**

We thank Shuang Liu and Arnab Kar for early discussions. KC and JI thank ONR under N00014-16-1-261, UC Lab Fees under LFR 18-548554 and NSF under 1253942 and 1804829 for support. AM was supported by the National Science Foundation under grants 1253327, 1408982; and by DARPA and SPAWAR under contract N66001-15-C-4067.

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
