[Supplementary Material]

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

 $\frac{1}{\sigma} \left( \frac{\sqrt{2\pi}}{\sigma} \right)^{\alpha-1}$ while the Renyi parameter grows with $(e^{\alpha/(2\sigma^2)})^{\alpha-1}$. Because $\frac{\sqrt{2\pi}}{\sigma} < e^{\alpha/(2\sigma^2)}$ for all $\sigma$, we conclude that modestly-sized values of $\alpha$ will cause the $(lin, Renyi)$ upper bound to fall below the Renyi parameter for all $\sigma$.

Figure 2a plots the $(lin, Renyi)$ upper bound, the exact $(lin, Renyi)$ parameter, and the Renyi-DP parameter as functions of $\alpha$ when $\sigma = 1$. We see the exact $(lin, Renyi)$ is always better than the Renyi-DP parameter, although the upper bound is worse for small $\alpha$. The upper bound overtakes the Renyi-DP parameter when $\alpha \approx 2.3$.

For the multidimensional Gaussian mechanism, the story is mostly the same. Note that the $(lin, Renyi)$ upper bound can be written as

$$\frac{1}{\alpha - 1} \log \left( 1 + 2^{d(\alpha-1)} \sqrt{\pi/2}^{\alpha-1} \left( \frac{\|v\|_\alpha}{\sigma} \right)^\alpha \right)$$

The Renyi parameter, on the other hand, is

$$\alpha \left( \frac{\|v\|_2}{\sigma} \right)^2$$

Notice that these are the same functions we looked at for the unidimensional case, but instead of $\frac{1}{\sigma}$, they are in terms of $\frac{\|v\|_\alpha}{\sigma}$ and $\frac{\|v\|_2}{\sigma}$, respectively. Indeed this is no accident, because the unidimensional cases assumed the $L_2$ sensitivity of the function was 1. However, when $\alpha > 2$, we have $\|v\|_\alpha < \|v\|_2$, so we can replace $\|v\|_\alpha$ with $\|v\|_2$, and the $(lin, Renyi)$ upper bound will only increase. But this still gives us the same conclusion as the unidimensional Gaussian mechanism, since now both parameters are functions of $\frac{\|v\|_2}{\sigma}$.

Finally, our multidimensional $(lin, Renyi)$ upper bound is slightly better than composing the Gaussian mechanism $d$ times which would result in a multiplicative factor of $d$.

Figure 2b contains plots of the exact $(poly, Renyi)$ parameters for degree 1,2, and 3 polynomials, as functions of $\alpha$ when $\sigma = 1$. As we expect, as the polynomial complexity increases, the $(poly, Renyi)$ parameters converge to the Renyi-DP parameter. This also provides an explanation for the counterintuitive observation that the $(poly, Renyi)$ parameters eventually decrease with $\alpha$. The polynomial function classes are too simple to distinguish the two distributions for large $\alpha$, but their ability to do so increases as the polynomial complexity increases.

# B  Post-Processing, Convexity, and Composition

*Proof.* (Of Theorem 1) It suffices to show Post-Processing Invariance for a restricted $f$-divergence. Let $A$ be a mechanism that maps a dataset $D$ into an output $x$ in an instance space $X$. Let $P = \Pr(A(D) = \cdot)$ and $Q = \Pr(A(D') = \cdot)$.

Suppose $g$ is a function in $\mathcal{G}$ which maps an $x \in X$ into a $y \in Y$ – that is $y = g(x)$. Let $P'$ and $Q'$ be the distributions induced on $Y$ by $P$ and $Q$ respectively when we map $x$ into $y$. To show post-processing, we need to show that $\mathcal{D}_f^{\mathcal{I}}(P', Q') \leq \mathcal{D}_f^{\mathcal{H}}(P, Q)$.

To see this, observe that:

$$
\begin{aligned}
\mathcal{D}_f^{\mathcal{I}}(P', Q') &= \sup_{i \in \mathcal{I}} \mathbb{E}_{P'}[i(y)] - \mathbb{E}_{Q'}[f^*(i(y))] \\
&= \sup_{i \in \mathcal{I}} \mathbb{E}_P[i \cdot g(x)] - \mathbb{E}_Q[f^*(i \cdot g(x))] \\
&\leq \sup_{h \in \mathcal{H}} \mathbb{E}_P[h(x)] - \mathbb{E}_Q[f^*(h(x))]
\end{aligned}
$$

where the first step follows because $y = g(x)$ and the second step follows because $i \cdot g \in \mathcal{H}$. The theorem follows from observing that the right hand side in the third line is exactly $\mathcal{D}_f^{\mathcal{H}}(P, Q)$.  □

*Proof.* (Of Theorem 2) To prove convexity, it suffices to show that
$$\mathcal{D}_\alpha^{\mathcal{H}}(M(D), M(D')) \leq \lambda \mathcal{D}_\alpha^{\mathcal{H}}(A(D), A(D')) + (1 - \lambda)\mathcal{D}_\alpha^{\mathcal{H}}(B(D), B(D'))$$
Observe that $\mathcal{D}_f^{\mathcal{H}}(M(D), M(D'))$ is equal to:

$$
\begin{aligned}
&= \sup_{h \in \mathcal{H}} \mathbb{E}_{x \sim M(D)}[h(x)] - \mathbb{E}_{x \sim M(D')}[f^*(h(x))] \\
&= \sup_{h \in \mathcal{H}} \mathbb{E}_{x \sim \lambda A(D) + (1-\lambda)B(D)}[h(x)] - \mathbb{E}_{x \sim \lambda A(D') + (1-\lambda)B(D')}[f^*(h(x))] \\
&= \sup_{h \in \mathcal{H}} \lambda \mathbb{E}_{x \sim A(D)}[h(x)] + (1-\lambda)\mathbb{E}_{x \sim B(D)}[h(x)] - \mathbb{E}_{x \sim \lambda A(D') + (1-\lambda)B(D')}[f^*(h(x))] \\
&= \sup_{h \in \mathcal{H}} \lambda \mathbb{E}_{x \sim A(D)}[h(x)] + (1-\lambda)\mathbb{E}_{x \sim B(D)}[h(x)] - \lambda \mathbb{E}_{x \sim A(D')}[f^*(h(x))] - (1-\lambda)\mathbb{E}_{x \sim B(D')}[f^*(h(x))] \\
&= \sup_{h \in \mathcal{H}} \lambda(\mathbb{E}_{x \sim A(D)}[h(x)] - \mathbb{E}_{x \sim A(D')}[f^*(h(x))]) + (1-\lambda)(\mathbb{E}_{x \sim B(D)}[h(x)] - \mathbb{E}_{x \sim B(D')}[f^*(h(x))]) \\
&\leq \lambda \cdot \sup_{h \in \mathcal{H}} \mathbb{E}_{x \sim A(D)}[h(x)] - \mathbb{E}_{x \sim A(D')}[f^*(h(x))] + (1-\lambda) \cdot \sup_{h \in \mathcal{H}} \mathbb{E}_{x \sim B(D)}[h(x)] - \mathbb{E}_{x \sim B(D')}[f^*(h(x))]
\end{aligned}
$$

where the second step follows from the fact that $M(D)$ is a mixture of $A(D)$ and $B(D)$ with mixing weights $[\lambda, 1 - \lambda]$, the third step is a property of mixture distributions, the fourth step from algebra, the fifth step from re-arrangement, and the last step from the observation that $\max_y f(y) + g(y) \leq \max_y f(y) + \max_y g(y)$. Observe that the last line is $\lambda \mathcal{D}_f^{\mathcal{H}}(A(D), A(D')) + (1 - \lambda)\mathcal{D}_f^{\mathcal{H}}(B(D), B(D'))$.  □

## B.1  Composition ($\mathcal{H}$-bounded Renyi, KL Privacy only)

*Proof.* (Of Theorem 4). Let $D$ and $D'$ be two datasets that differ in the private value of a single person, and let $(P_1, P_2) = (A(D), B(D))$ and $(Q_1, Q_2) = (A(D'), B(D'))$. Let $P$ be the product distribution $P_1 \otimes P_2$ and $Q$ be the product distribution $Q_1 \otimes Q_2$. Finally, let $a = \alpha^2 - \alpha$. By assumption, $\mathcal{D}_{R,\alpha}^{\mathcal{H}_i}(P_i, Q_i) \leq \epsilon_i$. Hence, we know $\mathcal{D}_\alpha^{\mathcal{H}_i}(P_i, Q_i) \leq \eta_i$ where $\eta_i = \frac{\exp(\epsilon_i(\alpha-1))-1}{a}$. Then,

$$
\begin{aligned}
\mathcal{D}_\alpha^{\mathcal{H}}(P, Q) &= \inf_{P'} \mathcal{D}_\alpha(P', Q) + \sup_{h \in \mathcal{H}} \mathbb{E}_P[h] - \mathbb{E}_{P'}[h] \\
&\leq \inf_{P' = P_1' \otimes P_2'} \mathcal{D}_\alpha(P', Q) + \sup_{h \in \mathcal{H}} \mathbb{E}_P[h] - \mathbb{E}_{P'}[h] \\
&= \inf_{P' = P_1' \otimes P_2'} a\mathcal{D}_\alpha(P_1', Q_1)\mathcal{D}_\alpha(P_2', Q_2) + \mathcal{D}_\alpha(P_1', Q_1) + \mathcal{D}_\alpha(P_2', Q_2) + \sup_{h \in \mathcal{H}} \mathbb{E}_P[h] - \mathbb{E}_{P'}[h] \\
&\leq \inf_{P' = P_1' \otimes P_2'} a\mathcal{D}_\alpha(P_1', Q_1)\mathcal{D}_\alpha(P_2', Q_2) + \mathcal{D}_\alpha(P_1', Q_1) + \mathcal{D}_\alpha(P_2', Q_2) \\
&\quad + \sup_{h_1 \in \mathcal{H}_1} \mathbb{E}_{P_1}[h_1] - \mathbb{E}_{P_1'}[h_1] + \sup_{h_2 \in \mathcal{H}_2} \mathbb{E}_{P_2}[h_2] - \mathbb{E}_{P_2'}[h_2]
\end{aligned}
$$

Here the first step follows from [21], the second step from restricting $P'$ to be a product distribution, third from the multiplicative property of $\alpha$-divergence for product distributions, fourth from the fact that we can split the sup of the product distributions into two parts. To simplify further, we use [21] again, this time on the assumptions:

$$\mathcal{D}_\alpha^{\mathcal{H}_1}(P_1, Q_1) = \inf_{P_1'} \mathcal{D}_\alpha(P_1', Q_1) + \sup_{h \in \mathcal{H}_1} \mathbb{E}_{P_1}[h] - \mathbb{E}_{P_1'}[h] \le \eta_1$$

$$\mathcal{D}_\alpha^{\mathcal{H}_2}(P_2, Q_2) = \inf_{P_2'} \mathcal{D}_\alpha(P_2', Q_2) + \sup_{h \in \mathcal{H}_2} \mathbb{E}_{P_2}[h] - \mathbb{E}_{P_2'}[h] \le \eta_2$$

Because $h$ contains constant functions, we know that $\sup_{h \in \mathcal{H}} \mathbb{E}_{P_i}[h] - \mathbb{E}_{P_i'}[h] \ge 0$, and thus

$$\mathcal{D}_\alpha^{\mathcal{H}_i}(P_i, Q_i) \le \eta_i$$

Continuing,

$$
\begin{aligned}
\mathcal{D}_\alpha^{\mathcal{H}}(P, Q) &\le \inf_{P' = P_1' \otimes P_2'} a\eta_1\eta_2 + \mathcal{D}_\alpha(P_1', Q_1) + \mathcal{D}_\alpha(P_2', Q_2) \\
&\quad + \sup_{h_1 \in \mathcal{H}_1} \mathbb{E}_{P_1}[h_1] - \mathbb{E}_{P_1'}[h_1] + \sup_{h_2 \in \mathcal{H}_2} \mathbb{E}_{P_2}[h_2] - \mathbb{E}_{P_2'}[h_2] \\
&\le a\eta_1\eta_2 + \inf_{P' = P_1' \otimes P_2'} \mathcal{D}_\alpha(P_1', Q_1) + \sup_{h_1 \in \mathcal{H}_1} \mathbb{E}_{P_1}[h_1] - \mathbb{E}_{P_1'}[h_1] \\
&\quad + \mathcal{D}_\alpha(P_2', Q_2) + \sup_{h_2 \in \mathcal{H}_2} \mathbb{E}_{P_2}[h_2] - \mathbb{E}_{P_2'}[h_2] \\
&\le a\eta_1\eta_2 + \eta_1 + \eta_2
\end{aligned}
$$

This means $\mathcal{D}_{R,\alpha}^{\mathcal{H}}(P, Q) \le \frac{1}{\alpha - 1} \log(a(a\eta_1\eta_2 + \eta_1 + \eta_2) + 1)$. We can simplify this:

$$
\begin{aligned}
\frac{1}{\alpha - 1} \log(a(a\eta_1\eta_2 + \eta_1 + \eta_2) + 1) &= \frac{1}{\alpha - 1}(\log(a\eta_1 + 1) + \log(a\eta_2 + 1)) \\
&= \epsilon_1 + \epsilon_2
\end{aligned}
$$

$\square$

*Proof.* (Of Theorem 3). Let $D$ and $D'$ be two datasets which differ in the value of a single row. Then, $D = (D_1, D_2)$, and we have two cases for $D'$: $D' = (D_1, D_2')$ or $(D_1', D_2)$ where the pairs $D_1, D_1'$ and $D_2, D_2'$ differ in one row. Suppose the first case is true. Then, $(P_1, P_2) = (A(D_1), B(D_2))$ and $(Q_1, Q_2) = (A(D_1), B(D_2'))$. Importantly, we have $P_1 = Q_1$. Then, letting $P = P_1 \otimes P_2$, $Q = Q_1 \otimes Q_2$, and $a = \alpha^2 - \alpha$,

$$
\begin{aligned}
\mathcal{D}_\alpha^{\mathcal{H}}(P, Q) &= \inf_{P'} \mathcal{D}_\alpha(P', Q) + \sup_{h \in \mathcal{H}} \mathbb{E}_P[h] - \mathbb{E}_{P'}[h] \\
&\le \inf_{P' = P_1 \otimes P_2'} \mathcal{D}_\alpha(P', Q) + \sup_{h \in \mathcal{H}} \mathbb{E}_P[h] - \mathbb{E}_{P'}[h] \\
&= \inf_{P' = P_1 \otimes P_2'} a\mathcal{D}_\alpha(P_1, Q_1)\mathcal{D}_\alpha(P_2', Q_2) + \mathcal{D}_\alpha(P_1, Q_1) + \mathcal{D}_\alpha(P_2', Q_2) + \sup_{h \in \mathcal{H}} \mathbb{E}_P[h] - \mathbb{E}_{P'}[h] \\
&= \inf_{P' = P_1 \otimes P_2'} \mathcal{D}_\alpha(P_2', Q_2) + \sup_{h \in \mathcal{H}} \mathbb{E}_P[h] - \mathbb{E}_{P'}[h] \\
&\le \inf_{P_2'} \mathcal{D}_\alpha(P_2', Q_2) + \sup_{h_1 \in \mathcal{H}_1} \mathbb{E}_{P_1}[h_1] - \mathbb{E}_{P_1}[h_1] + \sup_{h_2 \in \mathcal{H}_2} \mathbb{E}_{P_2}[h_2] - \mathbb{E}_{P_2'}[h_2] \\
&= \inf_{P_2'} \mathcal{D}_\alpha(P_2', Q_2) + \sup_{h_2 \in \mathcal{H}_2} \mathbb{E}_{P_2}[h_2] - \mathbb{E}_{P_2'}[h_2] \\
&= \mathcal{D}_\alpha^{\mathcal{H}_2}(P_2, Q_2)
\end{aligned}
$$

Here the first step follows from [21], the second step from restricting $P'$ to be a product distribution, third from the multiplicative property of $\alpha$-divergence for product distributions, fourth from the fact that $D(P_1, Q_1) = 0$ when $P_1 = Q_1$ for any divergence, fifth from splitting the sup into two parts, sixth from further cancellation. For the second case, where $D' = (D_1, D_2')$, we can prove $\mathcal{D}_\alpha^{\mathcal{H}}(P, Q) \le \mathcal{D}_\alpha^{\mathcal{H}_1}(P_1, Q_1)$ via a similar argument. With a simple transformation from $\alpha$ to $\alpha$-Renyi divergence, we obtain our result. $\square$

## C   Mechanisms

### C.1   KL, Unbounded

**Theorem 8** (Laplace Mechanism under KL). *Let $P$ and $Q$ be Laplace distributions with mean $0$ and $1$ and parameter $1/\epsilon$. Then,*

$$\mathbb{KL}(P, Q) = \epsilon - 1 + e^{-\epsilon}$$

*Proof.* We divide the real line into three intervals: $I_1 = [-\infty, 0]$, $I_2 = [0, 1]$, $I_3 = [1, \infty]$.

For any $x \in I_1$, $P(x)/Q(x) = e^{\epsilon}$, and $\Pr(I_1) = 1/2$ (under $P$). Therefore,

$$\mathbb{E}_P[\log(P/Q), x \in I_1] = \frac{1}{2}\epsilon$$

Similarly for any $x \in I_3$, $P(x)/Q(x) = e^{-\epsilon}$, and $\Pr(I_3)$ under $P$ is calculated as follows:

$$\Pr(I_3) = \int_1^{\infty} \frac{1}{2}\epsilon e^{-\epsilon x} dx = \frac{1}{2}e^{-\epsilon}$$

Therefore,

$$\mathbb{E}_P[\log(P/Q), x \in I_3] = -\frac{1}{2}\epsilon e^{-\epsilon}$$

We are now left with $I_2$. For any $x \in I_2$, we have $P(x)/Q(x) = e^{-\epsilon x}/e^{-\epsilon(1-x)} = e^{\epsilon(1-2x)}$. Therefore,

$$
\begin{aligned}
\mathbb{E}_P[\log(P/Q), x \in I_2] &= \int_0^1 \frac{1}{2}\epsilon^2(1 - 2x)e^{-\epsilon x} dx \\
&= \frac{1}{2}\epsilon^2 \left( \int_0^1 e^{-\epsilon x} dx - \int_0^1 2x e^{-\epsilon x} dx \right) \\
&= \frac{1}{2}\epsilon^2 \left( \frac{e^{-\epsilon x}}{-\epsilon}\bigg|_0^1 - \frac{2x e^{-\epsilon x}}{-\epsilon}\bigg|_0^1 + \int_0^1 \frac{2e^{-\epsilon x}}{-\epsilon}\bigg|_0^1 dx \right) \\
&= \frac{1}{2}\epsilon^2 \left( \frac{1 - e^{-\epsilon}}{\epsilon} + \frac{2e^{-\epsilon}}{\epsilon} - \int_0^1 2e^{-\epsilon x}\epsilon dx \right) \\
&= \frac{1}{2}\epsilon^2 \left( \frac{1 + e^{-\epsilon}}{\epsilon} - \frac{2e^{-\epsilon x}}{-\epsilon^2}\bigg|_0^1 \right) \\
&= \frac{1}{2}\epsilon^2 \left( \frac{1 + e^{-\epsilon}}{\epsilon} + \frac{2e^{-\epsilon} - 2}{\epsilon^2} \right)
\end{aligned}
$$

Summing up the three terms, we get:

$$\mathbb{E}_P[\log(P/Q)] = \frac{1}{2}\epsilon - \frac{1}{2}\epsilon e^{-\epsilon} + \frac{1}{2}\epsilon(1 + e^{-\epsilon}) + e^{-\epsilon} - 1 = \epsilon - 1 + e^{-\epsilon}$$

$\square$

The proof for the Gaussian Mechanism appears in Theorem 10.

### C.2   KL, Linear-Bounded

**Lemma 1.** *Let $\mathcal{X}$ be an instance space and $\phi$ be a vector of feature functions on $\mathcal{X}$ of length $M$. Let $lin$ be the class of functions:*

$$lin = \{a\phi(x) + b | a \in \mathbf{R}^M, b \in \mathbf{R}\}$$

*Then, for any two distributions $P$ and $Q$ on $\mathcal{X}$, we have:*

$$\mathbb{KL}^{\mathcal{H}}(P,Q) = \sup_{a \in \mathbf{R}^M} a^\top \mathbb{E}_{x \sim P}[\phi(x)] - \mathbb{E}_{x \sim Q}[e^{a^{\top-1}\phi(x)}]$$

$$\mathbb{KL}^{lin}(P,Q) = \sup_{a \in \mathbf{R}^M} a^\top \mathbb{E}_{x \sim P}[\phi(x)] - \log \mathbb{E}_{x \sim Q}[e^{a^\top \phi(x)}]$$

*Proof.* For KL-divergence, we have $f(x) = x \log(x)$. This means

$$f^*(s) = \sup_{x \in \mathbf{R}} xs - x \log x$$

The argument is maximized when $x = e^{s-1}$, so $f^*(s) = e^{s-1}$, and we obtain

$$\mathbb{KL}^{\mathcal{H}}(P,Q) = \sup_{h \in \mathcal{H}} \mathbb{E}_{x \sim P}[h(x)] - \mathbb{E}_{x \sim Q}[e^{h(x)-1}]$$

Now, we plug $lin$ into $\mathcal{H}$:

$$\mathbb{KL}^{lin}(P,Q) = \sup_{a \in \mathbf{R}^M, b \in \mathbb{E}} a^\top \mathbb{E}_{x \sim P}[\phi(x)] + b - \mathbb{E}_{x \sim Q}[e^{a^\top \phi(x)+b-1}]$$

Differentiating the objective with respect to $b$, we have that at the optimum:

$$1 - e^{b-1} \mathbb{E}_{x \sim Q}[e^{a^\top \phi(x)}] = 0,$$

which means that the optimum $b$ is equal to:

$$b = 1 - \log \mathbb{E}_{x \sim Q}[e^{a^\top \phi(x)}]$$

Plugging this in the objective, we get that:

$$
\begin{aligned}
\mathbb{KL}^{\mathcal{H}}(P,Q) &= \sup_{a \in \mathbf{R}^M} a^\top \mathbb{E}_{x \sim P}[\phi(x)] + 1 - \log \mathbb{E}_{x \sim Q}[e^{a^\top \phi(x)}] + (\mathbb{E}_{x \sim Q}[e^{a^\top \phi(x)}])^{-1} \mathbb{E}_{x \sim Q}[e^{a^\top \phi(x)}] \\
&= \sup_{a \in \mathbf{R}^M} a^\top \mathbb{E}_{x \sim P}[\phi(x)] - \log \mathbb{E}_{x \sim Q}[e^{a^\top \phi(x)}]
\end{aligned}
$$

The lemma follows. $\qquad\square$

**Theorem 9** (Laplace mechanism under $(lin, \mathbb{KL})$)**.** *Let $P = Lap(0, 1/\epsilon)$ and $Q = Lap(1, 1/\epsilon)$. Then,*

$$\mathbb{KL}^{lin}(P,Q) = \log\left(1 - \left(\frac{1 - \sqrt{1+\epsilon^2}}{\epsilon}\right)^2\right) + \sqrt{1+\epsilon^2} - 1$$

*Proof.* Recall that the density function of $P$ (resp. $Q$) is $\frac{\epsilon}{2} e^{-\epsilon|x|}$ (resp. $\frac{\epsilon}{2} e^{-\epsilon|x-1|}$). By Lemma 1, computing the linear KL is equivalent to solving the following problem:

$$\max_a a \mathbb{E}_{x \sim P}[x] - \log \mathbb{E}_{x \sim Q}[e^{ax}] = \max_a -\log\left(\frac{e^a}{1 - a^2/\epsilon^2}\right), a \in [-\epsilon, \epsilon] = \max_a \log(1 - a^2/\epsilon^2) - a, a \in [-\epsilon, \epsilon]$$

Note that the expression $\mathbb{E}_{x \sim Q}[e^{ax}]$ blows up to $\infty$ for $a \notin [-\epsilon, \epsilon]$, and hence the maximizer $a$ has to lie in $[-\epsilon, \epsilon]$. Taking the derivative and setting it to 0, we get:

$$\frac{-2a/\epsilon^2}{1 - a^2/\epsilon^2} - 1 = 0,$$

which, after some algebra, becomes the quadratic equation:

$$a^2 - 2a - \epsilon^2 = 0$$

The roots of this equation are: $a = 1 \pm \sqrt{1+\epsilon^2}$. The first root is more than $\epsilon$, and hence we choose $a = 1 - \sqrt{1+\epsilon^2}$ as the solution. Plugging this solution into the expression for $\mathbb{KL}^{lin}$, we get:

$$\mathbb{KL}^{lin}(P,Q) = \log\left(1 - \left(\frac{1 - \sqrt{1+\epsilon^2}}{\epsilon}\right)^2\right) + \sqrt{1+\epsilon^2} - 1$$

$\qquad\square$

**Theorem 10** (Gaussian mechanism under (lin)-KL)**.** *Let $P = \mathcal{N}(\mu_1, \sigma_1^2)$ and $Q = \mathcal{N}(\mu_2, \sigma_2^2)$. Then,*

$$\mathbb{KL}^{lin}(P, Q) = \frac{(\mu_1 - \mu_2)^2}{2\sigma_2^2}$$

*Proof.* By definition,

$$\mathbb{KL}^{lin}(P, Q) = \sup_a a\mu_1 - \log e^{a\mu_2 + a^2\sigma_2^2/2} = \sup_a a(\mu_1 - \mu_2) - \frac{1}{2}a^2\sigma_2^2$$

Differentiating wrt $a$ and setting the derivative to 0, the optimum is achieved at $a = (\mu_1 - \mu_2)/\sigma_2^2$, at which the optimal value is $(\mu_1 - \mu_2)^2/2\sigma_2^2$. $\qquad\square$

### C.3 Renyi, Unbounded

**Theorem 11** (Laplace Mechanism under $\alpha$-Renyi)**.** *Let $P$ and $Q$ be i.i.d. Laplace distributions in $d$ dimensions with mean 0 (resp. $v = (v_1, v_2, \ldots, v_d)$) and parameter $\frac{1}{\epsilon}$. Then,*

$$\mathcal{D}_{R,\alpha}(P, Q) = \frac{1}{\alpha - 1} \sum_{i=1}^{d} \log\left(\left(\frac{1}{2} + \frac{1}{4\alpha - 2}\right) e^{v_i(\alpha - 1)\epsilon} + \left(\frac{1}{2} - \frac{1}{4\alpha - 2}\right) e^{-v_i\alpha\epsilon}\right)$$

*Proof.* We first compute

$$\mathcal{D}_\alpha(P, Q) = \frac{1}{\alpha^2 - \alpha}\left(\int_{\mathbf{R}^d}\left(\frac{dP}{dQ}\right)^\alpha dQ - 1\right)$$

We write the integral as a product. Let $p_i$ be the p.d.f. for the $Lap(i, \frac{1}{\epsilon})$ distribution:

$$\int_{\mathbf{R}^d}\left(\frac{dP}{dQ}\right)^\alpha dQ = \int_{\mathbf{R}^d} P(x)^\alpha Q(x)^{1-\alpha}dx$$

$$= \int_{\mathbf{R}^d}\left(\prod_{i=1}^{d} p_0(x_i)^\alpha p_{v_i}(x_i)^{1-\alpha}\right) dx$$

$$= \prod_{i=1}^{d}\int_{\mathbf{R}} p_0(x_i)^\alpha p_{v_i}(x_i)^{1-\alpha}dx_i$$

We will compute each integral in the product individually. For the first case, suppose $v_i > 0$. We now split the real line into three regions: $I_1 = [\infty, 0]$, $I_2 = [0, v_i]$, and $I_3 = [v_i, \infty]$. Recall that $p_i(x) = \frac{\epsilon}{2}e^{-|x-i|\epsilon}$.

$$\int_{-\infty}^{0} p_0(x)^\alpha p_{v_i}(x)^{1-\alpha}dx = \frac{\epsilon}{2}\int_{-\infty}^{0}\left(\frac{e^{x\epsilon}}{e^{(x-v_i)\epsilon}}\right)^\alpha e^{(x-v_i)\epsilon}dx$$

$$= \frac{\epsilon}{2}\int_{-\infty}^{0} e^{v_i\alpha\epsilon - v_i\epsilon + x\epsilon}dx = \frac{1}{2}e^{v_i(\alpha-1)\epsilon}$$

$$\int_{0}^{v_i} p_0(x)^\alpha p_{v_i}(x)^{1-\alpha}dx = \frac{\epsilon}{2}\int_{0}^{v_i}\left(\frac{e^{-x\epsilon}}{e^{(x-v_i)\epsilon}}\right)^\alpha e^{(x-v_i)\epsilon}dx$$

$$= \frac{\epsilon}{2}\int_{0}^{v_i} e^{(1-2\alpha)x\epsilon + v_i(\alpha-1)\epsilon}dx = \frac{1}{2 - 4\alpha}e^{v_i(\alpha-1)\epsilon}(e^{v_i(1-2\alpha)\epsilon} - 1)$$

$$\int_{v_i}^{\infty} p_0(x)^\alpha p_{v_i}(x)^{1-\alpha} = \frac{\epsilon}{2}\int_{v_i}^{\infty}\left(\frac{e^{-x\epsilon}}{e^{(v_i-x)\epsilon}}\right)^\alpha e^{(v_i-x)\epsilon}dx$$

$$= \frac{\epsilon}{2}\int_{v_i}^{\infty} e^{-v_i\alpha\epsilon + v_i\epsilon - x\epsilon}dx = \frac{1}{2}e^{-v_i\alpha\epsilon}$$

$$\int_{\mathbf{R}} p_0(x)^\alpha p_{v_i}(x)^{1-\alpha}dx = \frac{1}{2}e^{v_i(\alpha-1)\epsilon} + \frac{1}{2}e^{-v_i\alpha\epsilon} + \frac{1}{2 - 4\alpha}(e^{-v_i\alpha\epsilon} - e^{v_i(\alpha-1)\epsilon})$$

$$= \left(\frac{1}{2} + \frac{1}{4\alpha - 2}\right) e^{v_i(\alpha-1)\epsilon} + \left(\frac{1}{2} - \frac{1}{4\alpha - 2}\right) e^{-v_i\alpha\epsilon}$$

When $v_i < 0$, we get the same answer, with $v_i$ replaced by $-v_i$. Let $F(x) = \left(\frac{1}{2} + \frac{1}{4\alpha-2}\right)e^{(\alpha-1)|x|} + \left(\frac{1}{2} - \frac{1}{4\alpha-2}\right)e^{-\alpha|x|}$. We can write

$$\mathcal{D}_\alpha(P,Q) = \frac{1}{\alpha^2 - \alpha}\left(\prod_{i=1}^{d} F(v_i\epsilon) - 1\right) \tag{5}$$

The expression for $\mathcal{D}_{R,\alpha}(P,Q)$ follows easily. $\qquad\square$

**Corollary 1.** *If $\alpha \geq 1$, then $\mathcal{D}_{R,\alpha}(P,Q) \geq \epsilon\|v\|_1$, where $v = (v_1, \ldots, v_d)$.*

*Proof.* When $\alpha > 1$, then $e^{(\alpha-1)|x|} > e^{-\alpha|x|}$. Thus, $F(x)$, defined above, is lower bounded by $\frac{1}{2}e^{(\alpha-1)|x|}$. Plugging into Equation (5), we get $\mathcal{D}_\alpha(P,Q) \geq \frac{1}{\alpha^2-\alpha}\left(e^{\epsilon(\alpha-1)\|v\|_1} - 1\right)$. The result for $\mathcal{D}_{R,\alpha}(P,Q)$ follows easily. $\qquad\square$

**Theorem 12** (Gaussian mechanism under $\alpha$-Renyi). *Let $P$ and $Q$ be i.i.d. Normal distributions in $d$ dimensions with mean $0$ (resp. $v = (v_1, v_2, \ldots, v_d)$) and variance $\sigma^2$. Then, $\mathcal{D}_{R,\alpha}(P,Q) = \frac{\alpha\|v\|_2^2}{2\sigma^2}$.*

*Proof.* We will compute

$$\mathcal{D}_\alpha(P,Q) = \frac{1}{\alpha^2 - \alpha}\left(\int_{\mathbf{R}^d}\left(\frac{dP}{dQ}\right)^\alpha dQ - 1\right)$$

Just like Theorem 11, we can write

$$\int_{\mathbf{R}^d}\left(\frac{dP}{dQ}\right)^\alpha dQ = \prod_{i=1}^{d}\int_{\mathbf{R}} p_0(x_i)^\alpha p_{v_i}(x_i)^{1-\alpha}dx_i$$

where $p_i(x)$ is the p.d.f. of $\mathcal{N}(i, \sigma^2)$. Therefore,

$$\int_{\mathbf{R}} p_0(x)^\alpha p_{v_i}(x)^{1-\alpha}dx = \frac{1}{\sqrt{2\pi\sigma^2}}\int_{-\infty}^{\infty}\left(\frac{e^{-x^2/(2\sigma^2)}}{e^{-(x-v_i)^2/(2\sigma^2)}}\right)^\alpha e^{-(x-v_i)^2/(2\sigma^2)}dx$$

$$= \frac{1}{\sqrt{2\pi\sigma^2}}\int_{-\infty}^{\infty} e^{(-(x+v_i(\alpha-1))^2 + v_i^2(\alpha-1)^2 + v_i^2(\alpha-1))/(2\sigma^2)}dx$$

$$= e^{v_i^2((\alpha-1)^2+(\alpha-1))/(2\sigma^2)} = e^{v_i^2(\alpha^2-\alpha)/(2\sigma^2)}$$

Therefore,

$$\mathcal{D}_\alpha(P,Q) = \frac{1}{\alpha^2 - \alpha}\left(\prod_{i=1}^{d} e^{v_i^2(\alpha^2-\alpha)/(2\sigma^2)} - 1\right) = \frac{1}{\alpha^2 - \alpha}(e^{\|v\|_2^2(\alpha^2-\alpha)/(2\sigma^2)} - 1)$$

The result for $\mathcal{D}_{R,\alpha}(P,Q)$ follows immediately. $\qquad\square$

### C.4 Renyi, Linear-Bounded

**Lemma 2** (Non-private Release). *Let $A$ be a mechanism such that there exist two datasets $D$ and $D'$ for which $A(D)$ and $A(D')$ are different point masses. Let $\mathcal{H}$ be a function class that contains linear functions. Then, $\mathcal{D}_{R,\alpha}(A(D), A(D')) = \infty$.*

*Proof.* Let $P$ denote the distribution of $A(D)$ and $Q$ denote the distribution of $A(D')$. If we show that $\mathcal{D}_\alpha^\mathcal{H}(P,Q) = \infty$, then certainly $\mathcal{D}_{R,\alpha}^\mathcal{H}(P,Q) = \infty$. Observe that $\mathcal{D}_\alpha^\mathcal{H}(P,Q) \geq \mathcal{D}_\alpha^{lin}(P,Q)$ by assumption. Hence, we are done if we show that $\mathcal{D}_\alpha^{lin}(P,Q) = \infty$.

$$\mathcal{D}_\alpha^{lin}(P,Q) \geq \sup_{a,b\in\mathbf{R}} \mathbb{E}_{x\sim P}[ax + b] - C_\alpha\mathbb{E}_{x\sim Q}[|ax + b|^{\alpha/(\alpha-1)}]$$

Suppose $P$ and $Q$ are point masses at $x = p$ and $x = q$ respectively. Then, there exists an $a$ and a $b$ such that $ap + b > 0$ and $aq + b = 0$. Pick any $\lambda > 0$. Plugging into the above,

$$\mathcal{D}_\alpha^{lin}(P,Q) \geq \lambda(ap + b) - \lambda(aq + b) = \lambda(ap + b)$$

Observe that as $ap + b > 0$ and is fixed with $\lambda$, $\lambda(ap + b) \to \infty$ as $\lambda \to \infty$. The lemma follows. $\qquad\square$

**Lemma 3.** *Let $\mathcal{X}$ be an instance space and $\phi$ be a vector of feature functions on $\mathcal{X}$ of length $M$. Then, for any two distributions $P$ and $Q$ on $\mathcal{X}$, we have:*

$$\mathcal{D}_\alpha^{\mathcal{H}}(P,Q) = \sup_{h \in \mathcal{H}} \mathbb{E}_{x \sim P}[h(x)] - C_\alpha \mathbb{E}_{x \sim Q}[|h(x)|^{\alpha/(\alpha-1)}] - \frac{1}{\alpha^2 - \alpha}$$

$$\mathcal{D}_\alpha^{lin}(P,Q) = \sup_{a \in \mathbf{R}^n, b \in \mathbf{R}} \mathbb{E}_{x \sim P}[a^T x + b] - C_\alpha \mathbb{E}_{x \sim Q}[|a^T x + b|^{\alpha/(\alpha-1)}] - \frac{1}{\alpha^2 - \alpha}$$

*where $C_\alpha = \frac{(\alpha-1)^{\alpha/(\alpha-1)}}{\alpha}$.*

*Proof.* We need to compute $f^*(s) = \sup_{x \in \mathbf{R}} xs - f(x)$ for $f = \frac{|x|^\alpha - 1}{\alpha^2 - \alpha}$ Setting the derivative of $f$ to zero, we get:

$$s - \text{sign}(x)\frac{|x|^{\alpha-1}}{\alpha-1} = 0 \implies x = (|s|(\alpha-1))^{1/(\alpha-1)}$$

Therefore,

$$f^*(s) = \frac{1}{\alpha}(|s|(\alpha-1))^{\alpha/(\alpha-1)} + \frac{1}{\alpha^2 - \alpha} = C_\alpha |s| + \frac{1}{\alpha^2 - \alpha}$$

This allows us to derive the expressions for $\mathcal{D}_\alpha^{lin}$ and $\mathcal{D}_\alpha^{\mathcal{H}}$ by plugging into

$$\mathcal{D}_\alpha^{\mathcal{H}}(P,Q) = \sup_{h \in \mathcal{H}} \mathbb{E}_{x \sim P}[h(x)] - \mathbb{E}_{x \sim Q}[f^*(h(x))]$$

$\square$

**Lemma 4.** *Suppose $P$ is a $d$-dimensional r.v. such that $\text{sign}(\mathbb{E}_{x \sim P}[x]) = (e_1, e_2, \ldots e_d)$ and $Q$ is a $d$-dimemsional r.v. with independent coordinates and marginals symmetric about 0. Then, the variational form of $\mathcal{D}_\alpha^{lin}(P,Q)$ can be written as*

$$\sup_{a \in \mathbf{R}^d, \text{sign}(a_i)=e_i, b \in \mathbf{R}, b \geq 0} \mathbb{E}_{x \sim P}[ax] + b - C_\alpha \mathbb{E}_{x \sim Q}[|ax + b|^{\alpha/(\alpha-1)}] - \frac{1}{\alpha^2 - \alpha}$$

*where $C_\alpha = \frac{(\alpha-1)^{\alpha/(\alpha-1)}}{\alpha}$.*

*Proof.* By Lemma 3,

$$\mathcal{D}_\alpha^{lin}(P,Q) = \sup_{a \in \mathbf{R}^d, b \in \mathbf{R}} \mathbb{E}_{x \sim P}[a^T x] + b - C_\alpha \mathbb{E}_{x \sim Q}[|a^T x + b|^{\alpha/(\alpha-1)}] - \frac{1}{\alpha^2 - \alpha} \qquad (6)$$

Let $a = (a_1, a_2, \ldots, a_d)$. The distribution of $|a^T Q + b|$ is determined only by the magnitude of each $a_i$, not its sign, because of the symmetry of $Q$. The sign of $b$ does not matter, either, as $|a^T Q - b| = |(-a)^T Q + b| = |a^T Q + b|$. Therefore, (6) achieves its maximum value when $\text{sign}(a_i) = e_i$ and $b > 0$, and we are done. $\square$

**Lemma 5.** *The function $f(a) = a - Xa^{\alpha/(\alpha-1)}$ has a global maximum of*

$$\frac{(\alpha-1)^{\alpha-1}}{\alpha^\alpha X^{\alpha-1}}$$

*Proof.* We observe that $f(a)$ is concave down over all real numbers. Its derivative vanishes when $a = \frac{(\alpha-1)^{\alpha-1}}{\alpha^{\alpha-1}X^{\alpha-1}}$, and because of the first observation, this is the global maximum. This is equal to

$$\frac{(\alpha-1)^{\alpha-1}}{\alpha^{\alpha-1}X^{\alpha-1}} - \frac{(\alpha-1)^\alpha}{\alpha^\alpha X^{\alpha-1}} = \frac{(\alpha-1)^{\alpha-1}}{\alpha^\alpha X^{\alpha-1}}$$

$\square$

**Theorem 13.** *(Symmetric Distributions under lin $\alpha$-Renyi): Suppose $P = X$ and $Q = X + v$ where $X$ is a $d$-dimensional r.v. consisting of $d$ i.i.d samples from an underlying distribution $Y$, $v \in \mathbf{R}^d$, and $Y$ is symmetric around 0 such that $\mathbb{E}[|Y|^{\alpha/(\alpha-1)}] = K$. Then,*

$$\mathcal{D}_{R,\alpha}^{lin}(P,Q) \leq \frac{1}{\alpha - 1} \log\left(1 + \frac{\|v\|_\alpha^\alpha}{(0.5^d K)^{\alpha-1}}\right)$$

*Proof.* We apply Lemma 4. For simplicity, let $A = \frac{\alpha}{\alpha-1}$. Then,

$$\mathcal{D}_\alpha^{lin}(P,Q) = \sup_{a \in \mathbf{R}^d, \text{sign}(a_i)=\text{sign}(v_i), b \in \mathbf{R}, b \geq 0} a^T v + b - C_\alpha \mathbb{E}_{x \sim Q}[|a^T x + b|^A] - \frac{1}{\alpha^2 - \alpha} \quad (7)$$

Because $Q$ is symmetric, we can write

$$\mathbb{E}_{x \sim Q}[|a^T x + b|^A] \geq \frac{1}{2^d} \mathbb{E}_{x \sim Q|\text{sign}(x)=\text{sign}(a_i)}[|a^T x + b|^A]$$

$$\geq \frac{1}{2^d} \mathbb{E}_{x \sim Q|\text{sign}(x_i)=\text{sign}(a_i)}\left[\sum_{i=1}^d (a_i x_i)^A\right] + b^A$$

$$= \frac{1}{2^d} \sum_{i=1}^d \mathbb{E}_{x_i \sim Y|\text{sign}(x_i)=\text{sign}(a_i)}[(a_i x_i)^A] + b^A$$

$$= \frac{1}{2^d} \sum_{i=1}^d |a_i|^A \mathbb{E}_{x_i \sim Y|\text{sign}(x_i)=\text{sign}(a_i)}[|x_i|^A] + b^A$$

$$= \frac{1}{2^d} \sum_{i=1}^d |a_i|^A K + b^A$$

Here, the first step comes from discarding the parts of the expectation where $\text{sign}(a_i) \neq \text{sign}(x)$ which is possible because the argument of the expectaion is always positive. Finally, the set that remains has measure measure $\frac{1}{2^d}$, so we normalize accordingly. The second step comes from the fact that $|a^T x + b|^A > \sum_{i=1}^d (a_i x_i)^A + b^A$ when $b, a_i x_i > 0$; the third comes from linearity of expecation; the fourth from the fact that $a_i$ and $x_i$ have the same sign; and the fifth from the fact that $Y$ is symmetric. We can now plug into (7) and simplify.

$$\mathcal{D}_\alpha^{lin}(P,Q) \leq \sup_{a \in \mathbf{R}^d, \text{sign}(a_i)=\text{sign}(v_i), b \in \mathbf{R}, b \geq 0} a^T v + b - C_\alpha \frac{1}{2^d} \sum_{i=1}^d |a_i|^A K - C_\alpha b^A - \frac{1}{\alpha^2 - \alpha}$$

$$\leq \sum_{i=1}^d \sup_{a_i \in \mathbf{R}, \text{sign}(a_i)=\text{sign}(v_i)} a_i v_i - \frac{C_\alpha K}{2^d} |a_i|^A + \sup_{b \in \mathbf{R}, b > 0} b - C_\alpha b^A - \frac{1}{\alpha^2 - \alpha}$$

$$\leq \sum_{i=1}^d |v_i| \sup_{a_i \in \mathbf{R}, a_i > 0} a_i - \frac{C_\alpha K}{2^d |v_i|} a_i^A + \sup_{b \in \mathbf{R}, b > 0} b - C_\alpha b^A - \frac{1}{\alpha^2 - \alpha}$$

$$= \sum_{i=1}^d |v_i| \frac{(\alpha-1)^{\alpha-1}}{\alpha^\alpha} \frac{|v_i|^{\alpha-1}(2^d)^{\alpha-1}}{C_\alpha^{\alpha-1} K^{\alpha-1}} + \frac{(\alpha-1)^{\alpha-1}}{\alpha^\alpha C_\alpha^{\alpha-1}} - \frac{1}{\alpha^2 - \alpha}$$

$$= \sum_{i=1}^d |v_i| \frac{1}{\alpha^2 - \alpha} \frac{|v_i|^{\alpha-1}(2^d)^{\alpha-1}}{K^{\alpha-1}} + \frac{1}{\alpha^2 - \alpha} - \frac{1}{\alpha^2 - \alpha}$$

$$= \frac{2^{d(\alpha-1)} \|v\|_\alpha^\alpha}{(\alpha^2 - \alpha) K^{\alpha-1}}$$

Here, the second line comes from the fact that a sup of a sum is at most the sum of sups, the third is from pulling out $v_i$ from the sup and being careful about signs, the fourth from applying Lemma 5, the fifth from plugging in for the $C_\alpha$ term and simplification, and the sixth from writing the answer in terms of norms. $\qquad \square$

**Corollary 2.** *For all $\alpha > 2$, the following are true:*

$$\mathcal{D}_{R,\alpha}^{lin}(\mathcal{N}(0,\sigma^2 I_d), \mathcal{N}(v,\sigma^2 I_d)) \leq \frac{1}{\alpha-1}\log\left(1 + \frac{\|v\|_\alpha^\alpha}{(0.5^d \times \sqrt{2/\pi})^{\alpha-1}\sigma^\alpha}\right)$$

$$\mathcal{D}_{R,\alpha}^{lin}(Lap_d(0,1/\epsilon), Lap_d(v,1/\epsilon)) \leq \frac{1}{\alpha-1}\log\left(1 + \frac{\|v\|_\alpha^\alpha \epsilon^\alpha}{(0.5^d)^{\alpha-1}}\right)$$

*Proof.* When $\alpha \geq 2$, $\frac{\alpha}{\alpha-1}$ is between 1 and 2, a rather small range. If $Y = \mathcal{N}(0,\sigma^2)$, then

$$K = \sigma^{\alpha/(\alpha-1)}\mathbb{E}[|\tilde{Y}|^{\alpha/(\alpha-1)}] \geq \sigma^{\alpha/(\alpha-1)}\inf_{1\leq\gamma\leq 2}\mathbb{E}[|\tilde{Y}|^\gamma]$$

where $\tilde{Y} = \mathcal{N}(0,1)$. $\mathbb{E}[|\tilde{Y}|^\gamma]$ is minimized when $\gamma = 1$, and we get $K \geq \sigma^{\alpha/(\alpha-1)}\sqrt{\frac{2}{\pi}} \approx 0.79\sigma^{\alpha/(\alpha-1)}$. We then apply Theorem 13.

If $Y = Lap(0, \frac{1}{\epsilon})$, then

$$K = \epsilon^{\alpha/(\alpha-1)}\mathbb{E}[|\tilde{Y}|^{\alpha/(\alpha-1)}] \geq \frac{1}{\epsilon^{\alpha/(\alpha-1)}}\inf_{1\leq\gamma\leq 2}\mathbb{E}[|\tilde{Y}|^\gamma]$$

where $\tilde{Y} = Lap(0,1)$. $\mathbb{E}[|\tilde{Y}|^\gamma]$ is minimized when $\gamma = 1$, and we get $K \geq \frac{1}{\epsilon^{\alpha/(\alpha-1)}}$. We then apply Theorem 13. $\qquad\square$

*Proof.* (Of Theorem 5) Let $x = (x_1, x_2, \ldots, x_n)$, and the columns of $A$ be $(a_1, a_2, \ldots, a_n)$. Changing $D$ in one place results in a change by 1 in at most one $x_i$. Thus, $Ax = \sum_{i=1}^{n} x_i a_i$ has $L_1$ sensitivity $\|A\|_1$. We can use the multidimensional Laplace mechanism (Corollary 2) which allows us to release $\tilde{a} = Ax + \|A\|_1 Lap_s(0, 1/\epsilon)$ offering $\mathcal{H}$-bounded privacy with parameter

$$\frac{1}{\alpha-1}\log\left(1 + \|v\|_\alpha^\alpha 2^{s(\alpha-1)}\frac{\epsilon^\alpha}{\|v\|_1^\alpha}\right)$$

where $\mathcal{H}$ is the set of linear functions : $\mathbf{R}^s \to \mathbf{R}$. Because $\|v\|_\alpha \leq \|v\|_1$, we can simplify this to $\frac{1}{\alpha-1}\log(1 + 2^{(\alpha-1)e}\epsilon^\alpha)$. We let $\mathcal{G}$ be the set of linear functions $\mathbf{R}^s \to \mathbf{R}^d$, $\mathcal{H}$ the set of linear functions $\mathbf{R}^s \to \mathbf{R}$, and $\mathcal{I}$ the set of linear functions $\mathbf{R}^d \to \mathbf{R}$. Notice that $WA^\dagger \in \mathcal{G}$ and $i \circ g \in \mathcal{H}$ for all $i \in \mathcal{I}$, $g \in \mathcal{G}$. Relasing $M_A(W, x, \epsilon) = WA^\dagger\tilde{a}$ then satisfies $\mathcal{H}$- capacity bounded privacy by post-processing. $\qquad\square$

## D   Generalization

*Proof.* (Of Theorem 6) Let $S = \{x_1, \ldots, x_n\}$ where $x_i$ are drawn iid from an underlying data distribution $D$. Let $S_{i\to x}$ denote $S$ with its $i$-th element $x_i$ replaced by $x$. Then, we have:

$$\mathbb{E}_{S\sim D^n, M}\left(\frac{1}{n}\sum_{i=1}^n q_S(x_i) - \mathbb{E}_{x\sim D}[q_S(x)]\right) = \mathbb{E}_{S\sim D^n}\frac{1}{n}\sum_{i=1}^n \left(\mathbb{E}_M[q_S(x_i)] - \mathbb{E}_{x\sim D,M}[q_S(x)]\right)$$

$$= \mathbb{E}_{S\sim D^n}\frac{1}{n}\sum_{i=1}^n \left(\mathbb{E}_M[q_S(x_i)] - \mathbb{E}_{x\sim D,M}[q_{S_{i\to x}}(x_i)]\right)$$

Here the first step follows from algebra, and the second step follows from observing that when $S \sim D^n, x \sim D$, $q_S(x)$ has the same distribution as when $S \sim D^n, x \sim D$, $q_{S_{i\to x}}(x_i)$. We pick any $i$. The term:

$$\mathbb{E}_M[q_S(x_i)] - \mathbb{E}_M[q_{S_{i\to x}}(x_i)] = \mathbb{E}_{q\sim M(S)}[q(x_i)] - \mathbb{E}_{q\sim M(S_{i\to x})}[q(x_i)]$$

$$= \mathbb{E}_{q\sim M(S)}[h_{x_i}(q)] - \mathbb{E}_{q\sim M(S_{i\to x})}[h_{x_i}(q)]$$

$$\leq \sup_{h\in\mathcal{H}}\mathbb{E}_{q\sim M(S)}[h(q)] - \mathbb{E}_{q\sim M(S_{i\to x})}[h(q)]$$

$$\leq \epsilon$$

Here the first step follows from simplifying notation, the second from the definition of $h_{x_i}$, the third from the fact that $\mathcal{H}$ includes $h_{x_i}$ and the fourth from the fact that $\text{IPM}^\mathcal{H}(M(S), M(S_{i\to x})) \leq \epsilon$. The theorem follows from combining this with Theorem 7. $\qquad\square$

*Proof.* (Of Theorem 7) If $\mathcal{H}$ is translation invariant and convex, then, we can write the $\mathcal{H}$-restricted KL divergence between any two distributions $P$ and $Q$ as follows: [21, 9]

$$\mathbb{KL}^{\mathcal{H}}(P,Q) = \inf_{\tilde{P}} \mathbb{KL}(\tilde{P},Q) + \sup_{h\in\mathcal{H}} \mathbb{E}_{x\sim P}[h(x)] - \mathbb{E}_{x\sim\tilde{P}}[h(x)] \tag{8}$$

Let $P'$ be the $\tilde{P}$ that achieves the infimum in (8). Then, from Pinsker Inequality, the left hand side of (8) is at least:

$$\frac{1}{2}(TV(P',Q))^2 + \sup_{h\in\mathcal{H}} \mathbb{E}_{x\sim P}[h(x)] - \mathbb{E}_{x\sim P'}[h(x)]$$

Let $\mathcal{F}$ be the class of all functions with range $[-1,1]$; by definition of the total variation distance, and because $\mathcal{H} \subseteq \mathcal{F}$, we have:

$$\mathrm{IPM}^{\mathcal{F}}(P,Q) = 2TV(P,Q) \geq \mathrm{IPM}^{\mathcal{H}}(P,Q)$$

Therefore $\mathbb{KL}^{\mathcal{H}}(P,Q)$ is at least

$$
\begin{aligned}
&\geq \quad \frac{1}{8}(\mathrm{IPM}^{\mathcal{H}}(P',Q))^2 + \mathrm{IPM}^{\mathcal{H}}(P,P') \\
&\geq \quad \frac{1}{16}(\mathrm{IPM}^{\mathcal{H}}(P',Q))^2 + (\mathrm{IPM}^{\mathcal{H}}(P,P'))^2 \\
&\geq \quad \frac{1}{64}(\mathrm{IPM}^{\mathcal{H}}(P',Q) + \mathrm{IPM}^{\mathcal{H}}(P,P'))^2 \\
&\geq \quad \frac{1}{64}(\mathrm{IPM}^{\mathcal{H}}(P,Q))^2
\end{aligned}
$$

Here the first step follows from Lemma 6, and the second step because as the range of any $h$ is $[-1,1]$, $\mathrm{IPM}^{\mathcal{H}}(P,P') \leq 2$, and hence $\mathrm{IPM}^{\mathcal{H}}(P,P') \geq \frac{1}{2}(\mathrm{IPM}^{\mathcal{H}}(P,P'))^2$. The third step follows because for any $a$ and $b$, $\frac{a^2}{2} + b^2 \geq \frac{1}{8}(a+b)^2$, and the final step from the triangle inequality of IPMs. The theorem thus follows. $\quad\square$

**Lemma 6.** *Let $\mathcal{H}$ be a function class that is closed under negation. Then, for any two distributions $P$ and $P'$,*

$$IPM^{\mathcal{H}}(P,P') = \sup_{h\in\mathcal{H}} \mathbb{E}_{x\sim P}[h(x)] - \mathbb{E}_{x\sim P'}[h(x)]$$

*Proof.* Observe that $\mathrm{IPM}^{\mathcal{H}}(P,P') \geq \sup_{h\in\mathcal{H}} \mathbb{E}_{x\sim P}[h(x)] - \mathbb{E}_{x\sim P'}[h(x)]$ by definition.

Now let $h'$ be function in $\mathcal{H}$ that achieves the supremum in $\mathrm{IPM}^{\mathcal{H}}(P,P')$. If $\mathbb{E}_{x\sim P}[h'(x)] \geq \mathbb{E}_{x\sim P}[h'(x)]$, then

$$\sup_{h\in\mathcal{H}} \mathbb{E}_{x\sim P}[h(x)] - \mathbb{E}_{x\sim P'}[h(x)] \geq \mathbb{E}_{x\sim P}[h'(x)] - \mathbb{E}_{x\sim P'}[h'(x)] = \mathrm{IPM}^{\mathcal{H}}(P,P'),$$

If not, then, $\mathbb{E}_{x\sim P}[-h'(x)] \geq \mathbb{E}_{x\sim P}[-h'(x)]$, and

$$\sup_{h\in\mathcal{H}} \mathbb{E}_{x\sim P}[h(x)] - \mathbb{E}_{x\sim P'}[h(x)] \geq \mathbb{E}_{x\sim P}[-h'(x)] - \mathbb{E}_{x\sim P'}[-h'(x)] = \mathrm{IPM}^{\mathcal{H}}(P,P')$$

The lemma follows. $\quad\square$