[Reviews · NeurIPS 2019]

Reviewer 1



The paper is technically strong, but it has some weaknesses in terms of the motivation and nomenclature. Firstly, I am not sure that 'capacity-bounded' makes intuitive sense for this definition, as there appears to be no direct link to capacity-like properties of the underlying function classes. ** Section 3 It is a bit unclear why $\cal{H}$ corresponds to some kind of adversary assumptions. In particular the statement 120-121 seems vacuous. If it is intended to offer an explanation of why $\cal{H}$ is a good way to represent adverary capabilities it fails. For the example given, why wouldn't an adversary just be able to perform a simple hypothesis test for $P$ versus $Q$, rather than be restricted to (2) ? The definition itself is very interesting technically, but the connection to adversary capabilities is far from apparent. Is the implication of e.g. a linear $\cal{H} that an adversary would be calculating a linear distinguisher between two possible neighbouring datasets? Why couldn't the adversary just use simulation and rely on $\cal{H}$ instead? Why wouldn't the adversary have access to the output of the cb-DP algorithm? ** Section 4 The properties in Section 4 are essential for a new definition to be of potential use, so they are in a sense the core part of the paper. ** Section 5 This gives us a couple of simple mechanisms as examples. While not essential, this is a useful section, that would perhaps have been nicely complemented by a small experimental result. ** Section 6 Here we have a generalisation bound, as well as an inequality extension to the restricted setting, which is also of potential general use. ** Appendix: Proof of Theorem 2, l. 459, step 4: The concavity claim is a bit strange for step 4. Wouldn't it be the case that for $Z = \lambda A + (1 - \lambda) B$, \[ E_Z [f^*] = \lambda E_A [f^*]+ (1 - \lambda) E_B [f^*] \] Just as in the previous line? I fail to see either a logarithm or a term which involves the expectation of a convex combination, i.e. \[ E[\lambda f + (1 - \lambda) g] \leq \lambda E[f] + (1 - \lambda) E[g], \] so step 4 should be an equality?

Reviewer 2



I have read the rebuttal. I think my concerns around the benefits and the threat model still remain. While I think it is definitely interesting to investigate various relaxations of differential privacy, in this case I do not see the clear utility benefits of using this weaker capacity bounded differential privacy notion. --------------- The paper presents capacity bounded differential privacy – a relaxation of differential privacy against adversaries in restricted function classes. This definition satisfies standard privacy axioms (such as convexity, post-processing invariance, and composition), and in the case where the adversary is limited to linear class in some cases it permits mechanisms that have higher utility. I think the overall the idea of capacity bounded differential privacy is neat. My main concern with the paper is that the paper does not make it obviously clear the benefits of using this relaxed privacy definition. At this point it feels that the risks (in terms of possible privacy breach) of using this relaxation outweighs the potential benefits. Unless this issue is addressed it is unclear to evaluate the potential impact of this definition.

Reviewer 3



The paper is extremely well organized, building up towards the (reasonably complex) definition without getting bogged down in technical details. The new relaxation of differential privacy is based on the variational interpretation of f-divergence, where the set of distinguishers is restricted to a particular functional class. My main concern is that any definition is only as good as it captures any desirable properties of the resulting construction. In the security domain (of which privacy is a part), new definitions are justified by presenting a security threat model where they somehow capture the adversary's capabilities. What is the threat model where the privacy adversary is limited to a class like linear functions? The submission offers two answers to this question. First, it considers an adversary that is itself an ML model of some restricted concept class. Second, it hypothesizes that the analyst may be contractually bound to perform only certain classes of computations. The main problem here is that the definition - as the submission correctly observes - is not closed under post-processing. In other words, once the (ML or contractually bound) adversary does its computation, _its_ output can be observed and processed by someone else, without restrictions imposed by the definition. The new definition does lead to better parameters at the expense of a stricter security model. The submission analyzes two basic mechanisms, Laplace and Gaussian, and compares their parameters for the same target level of Renyi DP (RDP). According the plots, RDP against the restricted adversary _improves_ for higher orders. (The standard RDP is monotone in the order.) This is an illustration of how different, and tricky, the new definition is.

[Author Response · NeurIPS 2019]

We thank all reviewers for their feedback. We respond to the main concerns. We will also address all the minor points
raised in the final version of the paper.

**Motivation / Threat Model / Use Case (Reviewers 1, 2, 3)** The main contribution of our paper is to provide a
framework that cleanly models privacy against bounded adversaries, and allows **quantitative calculation of privacy**
**loss** against particular adversarial classes. This was previously unknown (except for the special case of computationally
bounded adversaries). Quantification of privacy loss is important because it allows a systems-designer to precisely
determine the kind of privacy-accuracy tradeoff that is offered by a release.

One setting where quantifying privacy for different adversaries makes sense is when data sharing is coupled with data
usage contracts (as mentioned in Section 1 of the paper). For example, an instance of the Laplace mechanism might
offer $\epsilon = 1$ in general but $\epsilon = 1.25$ to a certain class of adversaries. Quantifying this tradeoff allows (a) better decisions
in cases where we expect the adversaries to be bounded in what they can do – for example, automated adversaries or
adversaries under a data-usage contract – and (b) better design of data-usage contracts – eg, if the loss against quadratic
functions is much higher than linear, then we can only allow for linear functions.

R3 astutely observes that problems may arise if the adversary does not obey the data usage contract, or its output is
viewed by someone else. In this case, we will sacrifice the improved privacy guarantee, but if we use a differentially
private mechanism, then we can fall back to the original (weaker) differential privacy guarantee that holds for all
adversaries.

We will add this discussion to the introduction in the final version of the paper.

**Reviewer 1** "The only thing missing from the paper is a better sense of how adversary types can be linked to function
classes, perhaps through an extensive example."

A concrete example is an excellent suggestion; we will do this in the final version.

"Why couldn't the adversary just use simulation and rely on $\mathcal{H}$ instead? Why wouldn't the adversary have access to the
output of the cb-DP algorithm?"

To clarify, the adversary here can only compute certain functions (those in $\mathcal{H}$) on the output of the capacity bounded
differential privacy algorithm. The above discussion about data usage agreements is one setting where adversaries of
this form arise.

We will correct the error pointed out in the Appendix as well as add theorem numbers to our references where applicable.

**Reviewer 2** "At this point it feels that the risks (in terms of possible privacy breach) of using this relaxation outweighs
the potential benefits."

We emphasize that the main contribution of the paper is quantifying the risk for capacity bounded adversaries. As
discussed above, under a mechanism (like a data usage agreement), one can get adversaries to only postprocess the
outputs of a mechanism using restricted function classes to get a tighter privacy analysis. If the adversary deviates, one
could still fallback to the general (and weaker) DP guarantee.

**Reviewer 3** "The main problem here is that the definition - as the submission correctly observes - is not closed under
post-processing. In other words, once the (ML or contractually bound) adversary does its computation, _its_ output can
be observed and processed by someone else, without restrictions imposed by the definition."

Data use agreements usually restrict the outright release of data or its derivatives. This would ensure that the output of a
capacity bounded DP algorithm is not released to an adversary of a different class. That said, if this does happen in
violation of the data use agreement, we can still fall back on the (weaker) DP guarantee that applies to all adversaries.

We will qualify our statement on the Groce et. al. paper in Section 1, as was suggested. We will also add the suggested
details to the definitions section.

[Meta-Review · NeurIPS 2019]

Relaxing the differential privacy definition is not an easy task, and this work takes a step in an interesting direction. The authors should discuss the shortfalls pointed out by the reviewers in their final version.